

# Reliability and validity of neurobehavioral function on the Psychology Experimental Building Language test battery in young adults

Brian J. Piper[1,2,3], Shane T. Mueller[4], Alexander R. Geerken[1], Kyle L. Dixon[5], Gregory Kroliczak[6], Reid H.J. Olsen[3] and Jeremy K. Miller[1]

[1] Department of Psychology, Willamette University, Salem, OR, United States
[2] Department of Psychology, Bowdoin College, Bowdoin, ME, United States
[3] Department of Behavioral Neuroscience, Oregon Health Sciences University, Portland, OR, United States
[4] Department of Cognitive and Learning Sciences, Michigan Technological University, Houghton, MI, United States
[5] Department of Psychology, University of New Mexico, Albuquerque, NM, United States
[6] Action and Cognition Laboratory, Department of Social Sciences, Institute of Psychology, Adam Mickiewicz University in Poznan, Poznan, Poland

Corresponding author
Brian J. Piper, psy391@gmail.com, bpiper@bowdoin.edu

## ABSTRACT

**Background.** The Psychology Experiment Building Language (PEBL) software consists of over one-hundred computerized tests based on classic and novel cognitive neuropsychology and behavioral neurology measures. Although the PEBL tests are becoming more widely utilized, there is currently very limited information about the psychometric properties of these measures.

**Methods.** Study I examined inter-relationships among nine PEBL tests including indices of motor-function (Pursuit Rotor and Dexterity), attention (Test of Attentional Vigilance and Time-Wall), working memory (Digit Span Forward), and executive-function (PEBL Trail Making Test, Berg/Wisconsin Card Sorting Test, Iowa Gambling Test, and Mental Rotation) in a normative sample ($N = 189$, ages 18–22). Study II evaluated test–retest reliability with a two-week interest interval between administrations in a separate sample ($N = 79$, ages 18–22).

**Results.** Moderate intra-test, but low inter-test, correlations were observed and ceiling/floor effects were uncommon. Sex differences were identified on the Pursuit Rotor (Cohen's $d = 0.89$) and Mental Rotation ($d = 0.31$) tests. The correlation between the test and retest was high for tests of motor learning (Pursuit Rotor time on target $r = .86$) and attention (Test of Attentional Vigilance response time $r = .79$), intermediate for memory (digit span $r = .63$) but lower for the executive function indices (Wisconsin/Berg Card Sorting Test perseverative errors $= .45$, Tower of London moves $= .15$). Significant practice effects were identified on several indices of executive function.

**Conclusions.** These results are broadly supportive of the reliability and validity of individual PEBL tests in this sample. These findings indicate that the freely downloadable, open-source PEBL battery (http://pebl.sourceforge.net) is a versatile research tool to study individual differences in neurocognitive performance.

# INTRODUCTION

A large collection of classic tests from the behavioral neurology and cognitive psychology fields have been computerized and made available (http://pebl.sf.net). This Psychology Experiment Building Language (PEBL) (*Mueller, 2010*; *Mueller, 2014a*; *Mueller, 2014b*; *Mueller & Piper, 2014*) has been downloaded over 168,000 times with 73% of downloads by users located outside of the United States, and used in scores of published manuscripts (e.g., *Barrett & Gonzalez-Lima, 2013*; *Danckert et al., 2012*; *Fox et al., 2013*; *González-Giraldo et al., 2014*; *González-Giraldo et al., 2015a*; *González-Giraldo et al., in press*; *Piper, 2011*; *Piper et al., 2012*; *Premkumar et al., 2013*; *Wardle et al., 2013*; Table S1). The growth in PEBL use is likely due to three factors. First, PEBL is free while other similar programs (*Robbins et al., 1994*) have costs that preclude use by all but the largest laboratories and are beyond the capacities of the majority of investigators in developing countries. Second, PEBL is open-source software and therefore the computational operations are more transparent than may be found with proprietary measures. Third, the distributors of some commercial tests restrict test availability to those who have completed specific coursework whereas PEBL is available to anyone with an internet connection. This investigation reports on the use of nine PEBL measures including convergent and divergent validity (Study I) and test–retest reliability (Study II). These measures were selected because of their distinct neuroanatomical substrates (Fig. 1) and because they measure theoretically important domains (Table S2), are practical to administer, and are commonly employed in earlier biobehavioral investigations (*Mueller & Piper, 2014*). A brief history of the more commonly utilized of these tests is provided below.

## Digit span

The origins of digit span, an extremely simple test in which strings of numbers of increasing length are presented and must be repeated back to the experimenter, are ambiguous but procedures that are analogous to what are frequently employed today date back at least as far as the pioneering developmental studies of Alfred Binet (*Richardson, 2007*). Although digit span is frequently described as an index of working memory, the importance of attention for optimal performance should not be underestimated (*Lezak et al., 2012*).

## Pursuit Rotor

The rotary pursuit test measures motor performance by using a stylus to track a target that moves clockwise at a fixed rate (*Ammons, Alprin & Ammons, 1955*). Procedural learning deficits using the rotary pursuit have been shown among patients with Huntington's (*Schmidtke et al., 2002*). As a result of the wide-spread use of the rotary pursuit in experimental psychology laboratories, a computerized version was developed. Unfortunately, this version could only generate linear target paths due to technical limitations at that time (*Willingham, Hollier & Joseph, 1995*). The PEBL Pursuit Rotor is a more faithful version

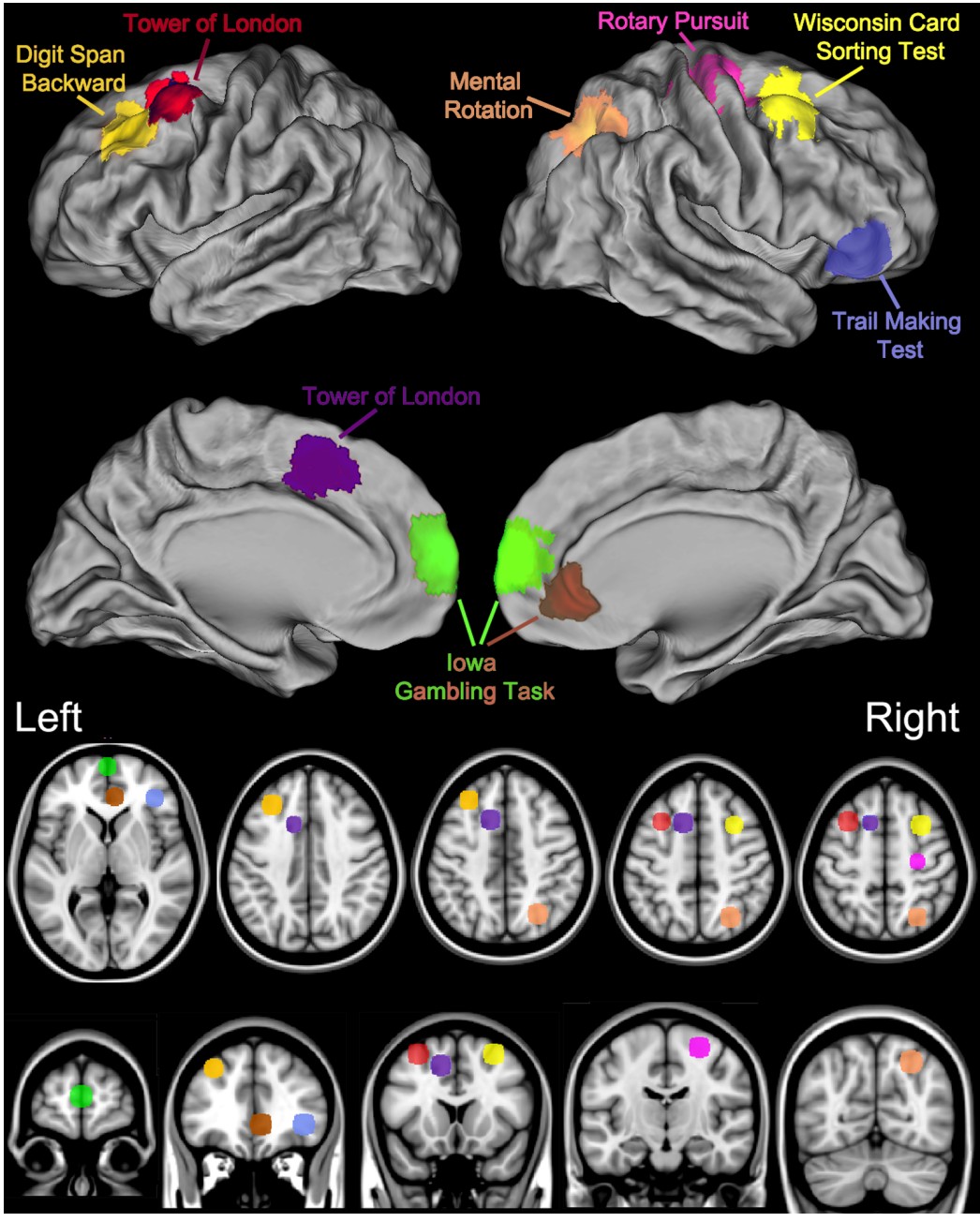

**Figure 1** Key brain areas as identified by neuroimaging and lesion studies and the corresponding Psychology Experiment Building Language Tests.

of the original rotary pursuit distributed by Lafayette instruments. Importantly, prior computer experience does account for a small portion of the variance in time on target (*Piper, 2011*).

## Wisconsin Card Sorting Test

The Wisconsin Card Sorting Test is a classic neuropsychological measure of cognitive flexibility and was originally developed at the University of Wisconsin following WWII

by *Grant & Berg (1948)*. In the original version, participants sorted physical cards into piles and determined the underlying classification principle by trial and error. Once consistent correct matching was achieved, the principle would be changed. The subsequent development of a computerized version of this complex task made for both more efficient use of the participants time and automated scoring (*Lezak et al., 2012*). Another key discovery was that 64 cards could be used instead of 128 (*Axelrod, Woodard & Henry, 1992*; *Fox et al., 2013*).

## Trail Making Test

The Trail Making Test is another of the oldest and most commonly employed neurobehavioral measures (*Lezak et al., 2012*). The Trail Making Test is typically thought to measure visual attention, mental flexibility, and executive functioning. The Trail Making Test was contained in the Army General Classification Test, a precursor of the Armed Services Vocational Aptitude Battery used by the United States military. The Trail Making Test involves connecting dots arranged in a numbered sequence in ascending order (Part A) or numbers and letters that alternate (Part B). Traditionally, performance on the Trail Making Test has been timed with a stop-watch and the experimenter has to redirect the participant when they make an error. Unlike the Halstead–Reitan Trail Making Test (*Gaudino, Geisler & Squires, 1995*), the path length is equal in Parts A and B of the PEBL Trail Making Test. Behavior on the Trail Making Test is sensitive to wide variety of insults including alcoholism (*Chanraud et al., 2009*).

## Mental rotation test

The mental rotation test has been an influential measure in cognitive psychology. Participants must decide whether an image is rotated in space and there is a linear relationship between the angle of rotation and decision time. Males typically exhibit better performance on spatial ability tests with some evidence indicating that this robust sex difference (e.g., *Yasen et al., 2015*) is detectable at very young ages (*Linn & Petersen, 1985*; *Moore & Johnson, 2008*).

## Tower of London

The Tower of London requires planning and judgment to arrive at the most efficient solution and move colored beads from their initial position to a new set of predetermined or goal positions (*Shallice, 1982*). There are many variations on this "brain teaser" type task including different levels of difficulty and construction (wood versus computerized) (*Lezak et al., 2012*). An elevation in the number of moves to solve Tower of London type problems has been documented among patients with brain damage and schizophrenia (*Morris et al., 1995*; *Shallice, 1982*; *Carlin et al., 2000*).

## Iowa Gambling Task

The Iowa Gambling Task was developed to model real world decision making in a laboratory environment. Participants receive $2,000 to start and must maximize their profit by choosing cards from among four decks of which two typically result in a net gain (+$250) and two result in a net loss (−$250). Although the Iowa Gambling Task has been

employed with a wide range of neuropsychiatric disorders, identification of a condition that consistently shows an abnormality on this test has proved difficult with the possible exception of problem gamblers (*Buelow & Suhr, 2009*; *Power, Goodyear & Crockford, 2012*).

## Test of Variables of Attention

The Test of Variables of Attention is an index of vigilance and impulsivity in which the participant responds to a target but inhibits responses for non-target stimuli. Although continuous performance tests were intended to discriminate children with, and without, Attention Deficit Hyperactivity Disorder (*Greenberg & Waldman, 1993*), the Test of Variables of Attention and other similar instruments may have proven even more valuable in measuring attention as a general construct and more specifically in evaluating the efficacy of cognitive enhancing drugs (*Huang et al., 2007*).

Another feature of the PEBL battery is that the key brain structures for these classic tasks are reasonably well characterized based on both lesion studies and more recent neuroimaging investigations (Fig. 1). Importantly, as diffuse neural networks are responsible for complex behaviors and the notion of a single neuroanatomical area underlying performance on a test risks oversimplification, more comprehensive information can be found elsewhere (*Demakis, 2004*; *Gerton et al., 2004*; *Grafton et al., 1992*; *Hugdahl, Thomsen & Ersland, 2006*; *Jacobson et al., 2011*; *Kaneko et al., 2011*; *Rogalsky et al., 2012*; *Schall et al., 2013*; *Specht et al., 2009*; *Tana et al., 2010*; *Zacks, 2008*). Briefly, completing the Pursuit Rotor with the dominant (right) hand results in a pronounced increase in blood flow in the left/right primary motor cortex, right cerebellum, the supplementary motor area, and the left putamen (*Grafton et al., 1992*). Tasks that require sustained attention engage the anterior cingulate and the insula (*Tana et al., 2010*). Digit Span activates the left prefrontal cortex when examined with near-infrared spectroscopy (*Kaneko et al., 2011*). Whole brain comparison of Digit Span backward, relative for forward, using Positron Emission Tomography (PET) revealed blood flow elevations in the dorsal lateral prefrontal cortex, left intraparietal lobule, and in Broca's area (*Gerton et al., 2004*). The Mental Rotation Test results in a robust activation in the right intraparietal sulcus as well as in the frontal and inferotemporal cortex (*Jacobson et al., 2011*). Executive function measures like the Trail Making Test, Iowa Gambling Test, Tower of London, and Wisconsin Card Sorting Test have been adapted from their clinical neuropsychological roots to be appropriate in a neuroimaging environment. Part B of the Trail Making Test, relative to Part A, produces Blood Oxygen Level Dependent elevations in the inferior/middle frontal gyri (*Jacobson et al., 2011*). The left middle frontal gyrus and right cerebellar tonsils show Tower of London difficulty dependent activations as determined by both functional magnetic resonance imaging and PET. The left ventral medial prefrontal cortex is engaged during completion of the Iowa Gambling Task (*Schall et al., 2013*) although lesion studies have produced conflicting evidence regarding the importance of this structure (*Carlin et al., 2000*; *Shallice, 1982*). The Wisconsin Card Sorting Test is a highly cognitively demanding task which involves an extremely diffuse cortical network including the right middle frontal gyrus as well as the left and right parietal lobule (*Kaneko et al., 2011*).

Previously, performance on three of the most prevalent executive function tests including the Wisconsin (Berg) Card Sorting Test, Trail Making Test, and the Tower of London was determined in a lifespan (age 5–87) sample. This investigation identified the anticipated "U-shaped" association between age and performance on these PEBL tests (*Piper et al., 2012*). One objective of the present report was to extend upon this foundation in a young-adult population by further examining the utility of the three executive function indices as well as six other tests including one (Dexterity) that is completely novel and another (Time-Wall) that is relatively obscure. Each participant in Study I completed all nine measures so that score distributions and the inter-test correlations could be evaluated. This information is necessary because PEBL measures, particularly the indices of executive function (i.e., measures that involve the frontal cortex, *Lezak et al., 2012*), are becoming increasingly utilized. The non-PEBL versions of several tests (Tower of London, Mental Rotation Test, Trail Making Test, rotary pursuit, and digit span) are often conducted using non-computerized methodology (*Lezak et al., 2012*) so it is currently unclear whether prior data on convergent and discriminant validity will be applicable. Many young adults have extensive experience with computerized measures so it is also crucial to determine whether any measures have ceiling effects.

With the exception of a single pilot study (*Piper, 2012*), there is currently no information about the test–retest reliability of individual PEBL tests or the battery. This dearth of data is unfortunate because the PEBL tests have already been employed in repeated measures designs (*Barrett & Gonzalez-Lima, 2013*; *Premkumar et al., 2013*; *Wardle et al., 2013*) and additional information would aid in the interpretation of those findings. The consistency of measurement is captured by two complementary measures. The correlation between the test and the retest measures the relative consistency, and the effect size quantifies the absolute consistency in performance.

There is a vast literature on the reliability of non-PEBL tests (*Calamia, Markon & Tranel, 2013*; *Lezak et al., 2012*) and a few investigations with similar methodology or sample characteristics similar to this report provide some context for the present endeavor. College-students assessed on a computerized target tracking task showed a high correlation ($r = .75$) across sessions separated by two weeks (*Fillmore, 2003*). Strong correlations ($r > .70$) were also noted on several indices of the Test of Variables of Attention among children completing that vigilance measure with a nine-day inter-test interval (*Learck, Wallace & Fitzgerald, 2004*). Veterans in their late-20s exhibited an intermediate ($r = .52$) consistency across three sessions (one/week) of a computerized Digit Span forward (*Woods et al., 2010*). The percentage selections of the disadvantageous decks showed a moderate correlation ($r \geq .57$) when the Iowa Gambling Task was administered thrice on the same day (*Lejuez et al., 2005*) but limited information is available at longer intervals (*Buelow & Suhr, 2009*). The magnitude of practice effects appears to be task dependent with slight changes identified for the Digit Span forward (*Woods et al., 2010*) and the Test of Variables of Attention (*Learck, Wallace & Fitzgerald, 2004*) but pronounced improvements for the Iowa Gambling Task (*Bechara, Damasio & Damasio, 2000*; *Lejuez et al., 2005*). Executive function tasks that have a problem solving element may, once solved, have a limited

reliability (*Lowe & Rabbitt, 1998*). For example, the correlation of the first with the second 64-trials on the Berg Card Sorting Test was relatively low ($r = .31$) (*Fox et al., 2013*). In fact, the Wisconsin Card Sorting Test has been referred to as a "one shot test" (*Lezak et al., 2012*).

Two secondary objectives of this report are also noteworthy. First, these datasets provided an opportunity to identify any sex differences on the PEBL battery. As a general rule, males and females are more similar than dissimilar on most neurocognitive measures. However, as noted previously, the Mental Rotation Test provides a clear exception to this pattern (*Linn & Petersen, 1985*; *Moore & Johnson, 2008*). A robust male advantage was observed among children completing the PEBL Pursuit Rotor task (*Piper, 2011*) and similar sex differences have been identified with the non-computerized version (*Willingham, Hollier & Joseph, 1995*) of this test. However, sex differences were most pronounced only at older (81+) but not younger (21–80) ages on a computerized task with many similarities to the PEBL Pursuit Rotor (*Stirling et al., 2013*).

A final objective was to evaluate the different card sorting rules on the Berg Card Sorting Task. The PEBL version of the Wisconsin Card Sorting Task has been employed in over a dozen reports (e.g., *Danckert et al., 2012*; *Fox et al., 2013*; *Piper et al., 2012*; *Wardle et al., 2013*) and may be the most popular of the PEBL tests. Importantly, the Berg Card Sorting Test was programmed based on the definitions of perseverative responses and perseverative errors contained in *Berg*'s *1948* report. Alternatively, the Wisconsin Card Sorting Task distributed by Psychological Assessment Resources employs the subsequent definitions of Robert Heaton and colleagues (*1993*).

Overall, this report provides data from two separate cohorts regarding the validity and test–retest reliability of several PEBL measures. This psychometric information is fundamental for others that may be considering using these tests.

## MATERIALS AND METHODS

### Participants

Participants (Study I: $N = 189$, 60.3% Female, Age $= 18.9 \pm 1.0$; Study II: $N = 79$, 73.0% Female, Age $= 19.1 \pm 0.1$) were college students receiving course credit. The test sequence in Study I was as follows: written informed consent, Tapping, Pursuit Rotor, Time-Wall, Trail-Making Task, Digit-Span Forward, Berg Card Sorting Test, Mental Rotation, Iowa Gambling Task, Tower-of-London, Dexterity, and the Test of Attentional Vigilance. Due to hardware technical difficulties, data from the tapping motor speed test were unavailable. Half of these measures (Time-Wall, Trail-Making Test, Digit-Span, Mental Rotation and the Test of Attentional Vigilance) contain programming modifications relative to the PEBL battery 0.6 defaults and may be found at: https://github.com/stmueller/pebl-custom/tree/master/Piper-PeerJ-2015b. All neurobehavioral assessments were completed on one of eight desktop computers running Microsoft Windows. Each of these tests is described further below and screen shots including instructions are in the Fig. S1. The number of tests was slightly reduced to eight for Study II and the sequence was a written informed consent followed by Pursuit Rotor, Trail-Making Test, Digit-Span, Test of Attentional Vigilance, Tower-of-London, Iowa Gambling Task, and Time-Wall. As Study I identified

subtle sex differences on Mental Rotation, this test was not included in Study II. Similarly, Study I found that an abbreviated version of the TMT could be a substitute for a longer version so a two-trial version was employed in Study II. The interval between the test and retest was two weeks (mean = 14.4 ± 0.2 days, Min = 11, Max = 24). This inter-test interval could be employed to examine the effects of a cognitively enhancing drug. All procedures are consistent with the Declaration of Helinski and were approved by the Institutional Review Board of Willamette University.

## PEBL tests

Pursuit Rotor measures motor-learning and requires the participant to use the computer mouse to follow a moving target on four fifteen-second trials. The target follows a circular path (8 rotations per minute) and the time on target and error, the difference in pixels between the cursor and target, were recorded (*González-Giraldo et al., in press*; *Piper, 2011*).

Time-Wall is an attention and decision making task that involves assessing the time at which a target, moving vertically at a constant rate, will have traveled a fixed distance. The primary dependent measure is Inaccuracy, defined as the absolute value of the difference between the participant response time and the correct time divided by the correct time (Minimum = 0.00). The correct time ranged from 2.0 to 9.2 s with feedback ("Too short" or "Too long") provided after each of the ten trials (*Piper et al., 2012*).

The Trail-Making Test is an index of executive function test and assesses set-shifting. In Set A, the participant clicks on an ascending series of numbers (e.g., 1—2—3—4). In Set B, the participant alternates between numbers and letters (e.g., 1—A—2—B). The primary dependent measure from the five trials is the ratio of total time to complete B/A with lower values (closer to 1.0) indicative of better performance. Based on the findings of Study I with five trials, only the first two trials were completed in Study II.

In the PEBL default Digit Span forward, strings of numbers of increasing length starting with three were presented via headphones and displayed at a rate of one/second. Audio feedback (e.g., "Correct" or "Incorrect") was provided after each of three trials at each level of difficulty. The primary dependent measure was the number of trials completed correctly.

The Berg Card Sort Test measures cognitive flexibility and requires the participant to sort cards into one of four piles based on a rule (color, shape, number) that changes. Feedback ("correct!" or "incorrect") was displayed for 500 ms after each trial. This test differs somewhat from the version employed previously (*Fox et al., 2013*; *Piper et al., 2012*) in that the prior selections were displayed (Fig. S1E). The primary dependent measure is the percent of the 64 responses that were perseverative errors defined and coded according to the Heaton criteria (*Heaton et al., 1993*) although the number of categories completed and perseverative responses was also recorded.

In the PEBL Mental Rotation Test, the participant must decide whether two 2-dimensional images are identical or if one is a mirror image. There are a total of 64 trials with the angle of rotation varied in 45° increments (−135° to +180°). The percent correct and response time were the dependent measures.

In the PEBL Iowa Gambling Task, the four decks are labeled 1, 2, 3, and 4 rather than A, B, C, and D (*Buelow & Suhr, 2009*). The primary dependent measure was the $ at the end and response preference ((Deck 3 + Deck 4) − (Deck 1 + Deck 2)) with Decks 3 and 4 being advantageous and Decks 1 and 2 being disadvantageous. The response to feedback and the frequency different strategies were employed, e.g., payoff and then change piles (Win-Switch), lose money but continue with the same pile (Lose-Stay), was also documented.

In the Tower-of-London, the participant must form a plan in order to move colored disks, one at a time, to match a specified arrangement. The number of points to solve twelve problems (3 points/problem) and the average completion time/problem were recorded. Based on some indications of ceiling effects in Study I, Study II employed a more challenging version of this task (*Piper et al., 2012*) with the primary measure being moves and completion time as a secondary measure.

Dexterity is a recently developed test of fine motor function that consists of a circular coordinate plane with the center of the circle (demarcated by a thin black line) at x,y positions 0,0. The goal is to move the cursor (depicted as a colored ball) to a target located at various positions. Movement of the cursor is affected by a "noise" component complementing the directional input from the analog mouse to create the effect of interference or "jittering" motion. The effect is such that successful navigation of the coordinate plane using the mouse encounters resistance to purposeful direction, requiring continual adjustment by the participant to maintain the correct path to the target. Visual feedback is given by the use of a color system, wherein the cursor shifts gradually from green to red as proximity to the target becomes lesser. The task consists of 80 trials (10 per "noise" condition), ten seconds maximum in length, with preset noise factors (ranging in intensity) and target locations standardized for consistency between participants. A lack of input from the participant results in a gradual drift towards the center. At the conclusion of each trial, the cursor location is reset to the origin. Completion time and Moves were recorded with Moves defined as the change in the vector direction of the mouse while course correcting toward the target (Fig. S1H).

Finally, in the Test of Attentional Vigilance, participants are presented with "go/no-go" stimuli that they must either respond or inhibit their response. An abbreviated version (6 min) was employed. The primary dependent measures were the reaction time and the variability of reaction times. All PEBL source code from these studies is available at: https://github.com/stmueller/pebl-custom).

## Data analysis

All analyses were conducted using Systat, version 13.0 with figures prepared using Prism, version 6.03. Ceiling and floor effects were determined by examining score distributions for any measure with ≥5% of respondents scoring at the maximum or minimum of the obtainable range on that measure. As the PEBL default criteria for perseverative errors on the Berg Card Sorting Test is currently very different than that employed by *Heaton et al. (1993)* in the Wisconsin Card Sorting Test, secondary analyses were completed with each definition. Sex differences in Study I and the magnitude of practice effects (Study II) were

**Table 1** Performance on the Psychology Experimental Building Language (PEBL) battery including total time on target on the Pursuit Rotor (PR), Response Time (RT) and RT standard deviation (SD) on the Test of Attentional Vigilance (TOVA), B:A ratio on the Trail-Making Test (TMT), Tower of London (ToL), Perseverative Errors (PE) on the Wisconsin (Berg) Card Sorting Test (BCST), and Mental Rotation Test (MRT).

| | Min ($N$) | Max ($N$) | Mean | SEM | $N$ |
|---|---|---|---|---|---|
| A. Pursuit Rotor: time (sec) | 18.8 (1) | 56.3 (1) | 44.0 | 0.5 | 189 |
| B. Pursuit Rotor: error (pixels) | 50.5 (1) | 322.7 (1) | (1) 87.7 | 2.6 | 189 |
| C. Dexterity (sec) | 956.9 (1) | 7,276.8 (1) | 1,619.4 | 59.2 | 175 |
| D. Time-Wall (% inaccuracy) | 3.0 (1) | 53.0 (1) | 10.2 | 0.5 | 171 |
| E. Test of Attentional Vigilance: RT (ms) | 269 (1) | 495 (1) | 339.6 | 3.2 | 150 |
| F. Test of Attentional Vigilance: RT SD | 42 (1) | 288 (1) | 100.3 | 2.6 | 150 |
| G. Digit Span (points) | 7 (7) | 21 (3) | 13.5 | 0.3 | 148 |
| H. Trail Making Test (B:A) | 0.62 (1) | 2.10 (1) | 1.28 | 0.02 | 180 |
| I. Tower of London (points) | 12 (1) | 36 (6) | 29.0 | 0.3 | 182 |
| J. Tower of London (sec/trial) | 4.8 (1) | 32.0 (1) | 14.3 | 0.4 | 182 |
| K. Iowa Gambling Test ($) | −500 (1) | 4,500 (1) | 1,894 | 54 | 184 |
| L. Berg Card Sorting Test (% PE Heaton) | 3.1 (2) | 65.6 (1) | 11.0 | 0.5 | 173 |
| M. Berg Card Sorting Test (% PE Berg) | 0.0 (2) | 35.9 (1) | 12.9 | 0.5 | 174 |
| N. Mental Rotation Test (% correct) | 34.4 (1) | 100.0 (1) | 73.9 | 1.4 | 174 |
| O. Mental Rotation Test (ms) | 420.4 (1) | 5,381.6 (1) | 2,564.6 | 66.2 | 174 |

expressed in terms of Cohen's $d$ (e.g., (Absolute value (Mean$_{\text{Retest}}$ − Mean$_{\text{Test}}$)/SD$_{\text{Test}}$) with 0.2, 0.5, and 0.8 interpreted as small, medium, and large effect sizes. In Study II, correlation ($r$ and rho) and paired $t$-tests were calculated on the test and retest values. Test-retest correlations >0.7 were interpreted as acceptable (*Nunnally & Bernstein, 1994*) and <0.3 as unacceptable. The percent change was determined in order to facilitate comparison across measures.

# RESULTS

## Study I: normative behavior & inter-test associations

The nine PEBL tests may be organized into the following broad domains: motor function (Pursuit Rotor and Dexterity), Attention (Test of Attentional Vigilance and Time-Wall), Working-Memory (Digit Span), and Executive Functioning/Decision Making (Trail Making Test, Tower of London, Berg Card Sorting Test, Iowa Gambling Test, and the Mental Rotation Test). Table 1 shows that there were substantial individual differences in this sample. With the exception of the Tower of London (Maximum Possible Points = 36), no test showed evidence of ceiling or floor effects. The Berg criteria for coding perseverative responses resulted in a many more than the Heaton criteria (Mean$_{\text{Berg}}$ = 30.8 ± 6.9%, Mean$_{\text{Heaton}}$ = 11.9 ± 8.1%, $t(172) = 24.10, P < .0005$). The difference for perseverative errors was more subtle but still significant (Mean$_{\text{Berg}}$ = 12.9 ± 5.8%, Mean$_{\text{Heaton}}$ = 11.0 ± 6.4%, $t(172) = 3.79, P < .0005$) on the Berg Card Sorting Test.

Overall, sex differences were infrequent. On the Pursuit Rotor, the total time on target was greater in males (47.6 ± 6.0) than females (41.8 ± 7.0 s, $t(182) = 5.79, P < .0005$,

**Table 2** Spearman correlations between tests on the Psychology Experimental Building Language (PEBL) battery including Response Time (RT) and RT standard deviation (SD), Part B to Part A ratio on the Trail-Making Test; Perseverative Errors (PE) on the Berg Card Sorting Test coded according to the [B]Berg and [H]Heaton criteria. Correlations in **bold** are significant at $P \leq .05$, those in both ***bold and italics*** are significant at $P < .0005$.

| | A. | B. | C. | D. | E. | F. | G. | H. | I. | J. | K. | L. | M. |
|---|---|---|---|---|---|---|---|---|---|---|---|---|---|
| A. Pursuit Rotor: time (sec) | +1.00 | | | | | | | | | | | | |
| B. Pursuit Rotor: error | *−0.96* | +1.00 | | | | | | | | | | | |
| C. Dexterity (ms) | *−0.26* | **+0.25** | +1.00 | | | | | | | | | | |
| D. Time-Wall (inaccuracy) | *−0.32* | **+0.32** | +0.13 | +1.00 | | | | | | | | | |
| E. TOAV: RT (ms) | −0.13 | +0.14 | +0.12 | +0.13 | +1.00 | | | | | | | | |
| F. TOAV: RT SD | *−0.38* | **+0.38** | +0.18 | **+0.25** | *+0.46* | +1.00 | | | | | | | |
| G. Digit Span | +0.14 | **−0.18** | −0.10 | −0.08 | **−0.24** | −0.11 | +1.00 | | | | | | |
| H. Trail Making Test (B:A) | −0.16 | +0.16 | +0.06 | +0.09 | +0.12 | −0.04 | −0.01 | +1.00 | | | | | |
| I. Tower of London (points) | +0.15 | **−0.17** | −0.13 | −0.14 | −0.14 | −0.13 | +0.00 | **−0.27** | +1.00 | | | | |
| J. Iowa Gambling Test | +0.01 | −0.02 | −0.09 | −0.13 | +0.16 | +0.05 | +0.02 | −0.10 | −0.00 | +1.00 | | | |
| K. Berg Card Sorting Test (% PE[H]) | **−0.21** | *+0.27* | −0.03 | +0.12 | +0.15 | +0.07 | −0.12 | **+0.27** | **−0.34** | −0.07 | +1.00 | | |
| L. Berg Card Sorting Test (% PE[B]) | **−0.18** | **+0.22** | +0.02 | +0.07 | +0.06 | +0.06 | −0.02 | **+0.30** | −0.19 | −0.01 | *+0.72* | 1.00 | |
| L. Mental Rotation Test (% correct) | **+0.20** | **−0.20** | −0.07 | −0.04 | −0.08 | **−0.19** | +0.06 | −0.08 | **+0.29** | −0.04 | **−0.20** | **−0.18** | +1.00 |
| M. Mental Rotation Test (ms) | −0.09 | +0.05 | −0.07 | −0.01 | +0.12 | −0.05 | **+0.19** | +0.10 | +0.13 | +0.11 | +0.05 | +0.03 | +0.08 |

$d = 0.89$). Further analysis determined that target time in males was elevated by over 1,300 ms on each trial relative to females (Fig. 2A). On the Mental Rotation Test, there was no sex difference in the percent correct (Females $= 72.8 \pm 17.9\%$, Males $= 74.9\% \pm 19.4\%$, $t(168) = 0.47$). Decision time was increased by the angle and the number correct decreased as the rotation angle extended away from zero degrees in either direction (Fig. 2B). The sex difference (Males $= 2{,}377.8 \pm 795.8$, Females $= 2{,}638.0 \pm 863.3$) for overall response time was barely significant ($t(168) = 1.99$, $P < .05$, $d = 0.31$) with more pronounced group differences identified at specific angles (e.g., $−45°$, $d = 0.51$). Further, on the Iowa Gambling Test, total amount earned at the end of the game did not show a sex difference (Males $= \$1{,}928.95 \pm 707.99$, Females $= 1{,}858.02 \pm 755.47$, $t(180) = 0.64$, $P = .52$) but, following a loss, Males ($3.0 \pm 3.5$) were 73.1% more likely on their following choice to select again from the same deck (Females $= 1.7 \pm 2.8$, $t(136.6) = 2.86$, $P < .01$, $d = 0.41$).

Table 2 depicts the correlations among the tests. Generally, the association within measures on a single test was moderate to high (e.g., Pursuit Rotor, Test of Attentional Vigilance, Berg Card Sorting Test) whereas between tests Spearman rho values were typically lower. Lower performance on the Pursuit Rotor (i.e., higher Error) was associated with less attentional consistency (i.e., larger Test of Attentional Vigilance variability), longer times to complete Dexterity, more Perseverative Errors on the Berg Card Sorting Test, greater Time-Wall Inaccuracy, and lower Digit Span forward. There were also several correlations on the indices of executive function. Individuals that performed less well on the Trail Making Test (i.e., higher B to A ratios) scored lower on the Tower of London and the Berg Card Sorting Test. The correlation between Berg Card Sorting Test perseverative errors when coded according to the Heaton and default (Berg) criteria was moderately high. More correct Mental Rotation responses also corresponded with higher performance

### A. Pursuit rotor

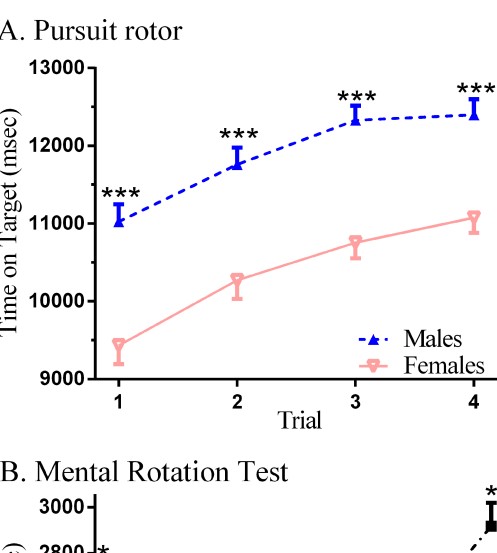

### B. Mental Rotation Test

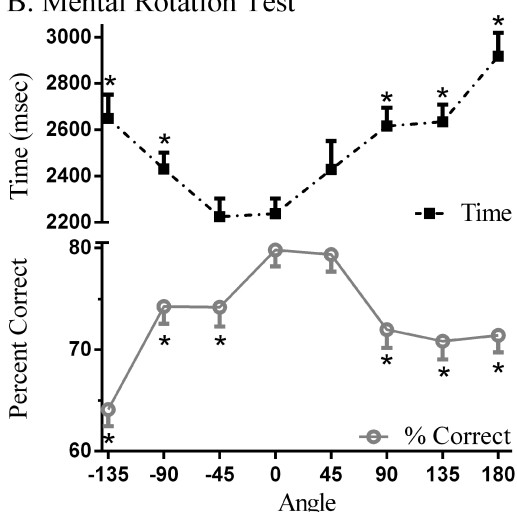

### C. Trail-Making Test

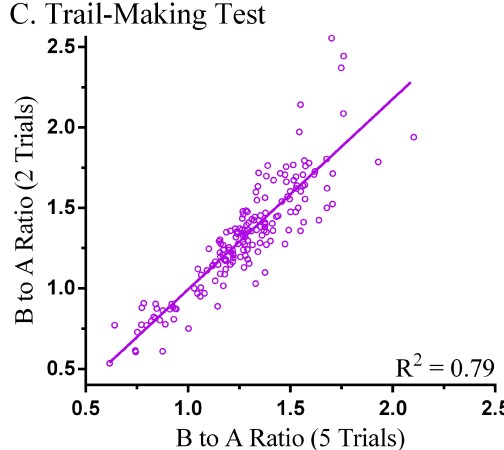

**Figure 2 Neurobehavioral performance on Psychology Experiment Building Language (PEBL) tests.** (A) Time on target on the Pursuit Rotor (***$P < .0005$ versus Females); (B) decision time and percent correct on the Mental Rotation (*$P < .0005$ versus Angle $= 0°$); (C) scatterplot of the ratio (Part B/Part A) of times to complete five versus two trials of the PEBL Trail-Making Test.

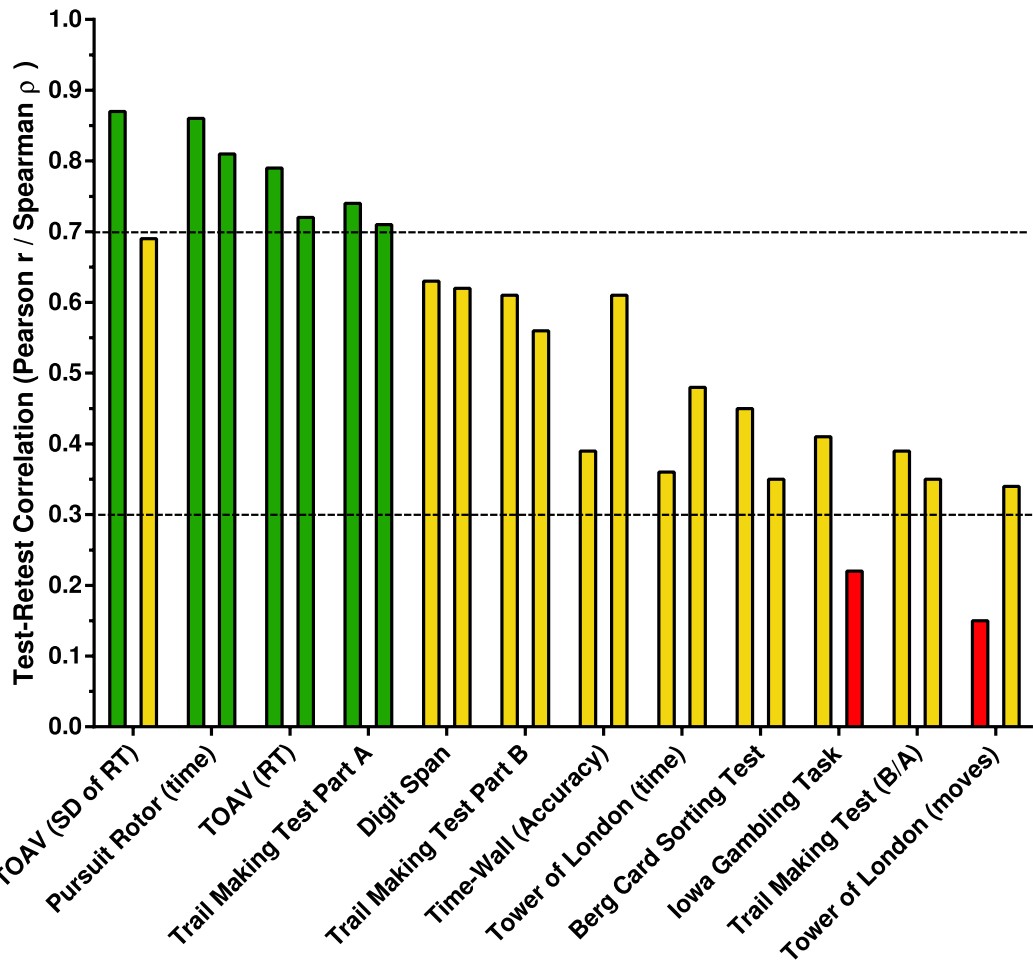

**Figure 3 Test-retest correlations ranked from highest to lowest.** For each Psychology Experiment Building Language Test, the Pearson *r* is listed first followed by the Spearman rho. Correlations ≥.7 are acceptable and below 0.3 as unacceptable. RT, Response Time; TOAV, Test of Attentional Vigilance; Trail Making Test Ratio of Completion times for Part B/Part A (B/A).

on the Tower of London. Also noteworthy, the B to A Ratio with all five trials showed a strong correspondence with only the first two Trail Making Test trials ($r_S(178) = +0.90$, $P < .0005$, Fig. 2C).

## Study II: test–retest reliability

Figure 3 shows the test–retest correlations ranked from highest to lowest. Spearman and Pearson correlations ≥0.7 were interpreted as acceptable, ≥0.3 and <.7 as intermediate, and below 0.3 as unacceptable. Acceptable correlations were identified on the Pursuit Rotor and the Test of Attentional Vigilance. Digit Span, Time-Wall, and most measures on the Berg Card Sorting Test were intermediate. Select correlations were below the acceptability cut-off for the Iowa Gambling Task and the Tower of London. The reliability of secondary measures is also listed on Table S3. Most notably, reliability coefficients on the Berg Card Sorting Test were equivalent for perseverative errors with the Berg and Heaton definitions.

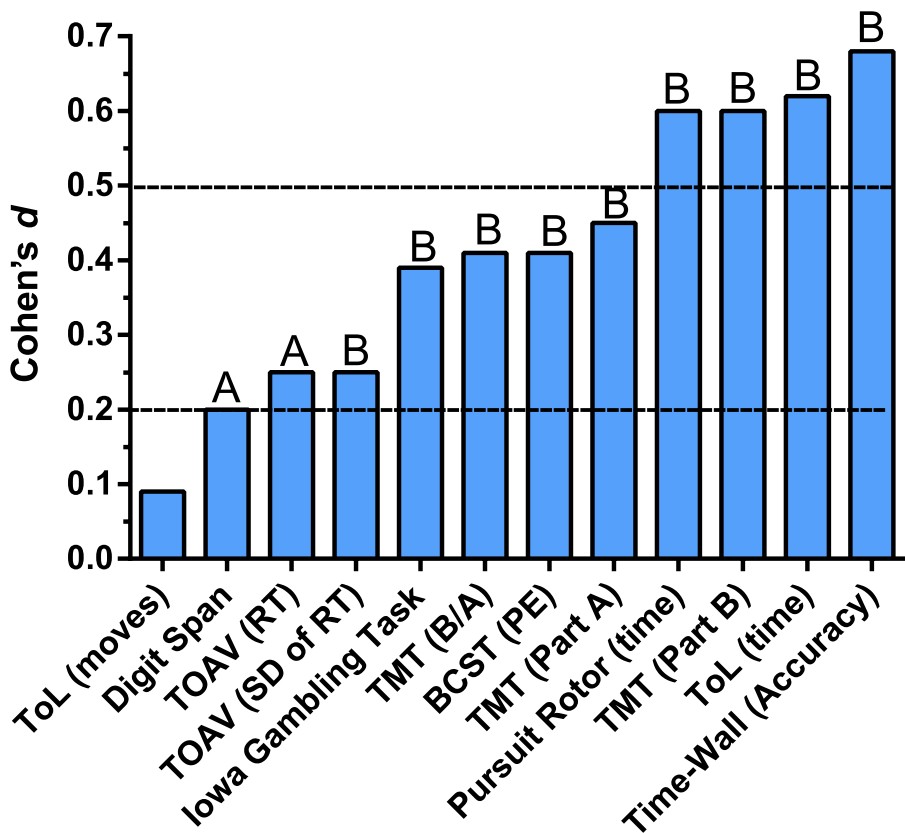

**Figure 4 Change from the test to the retest, expressed as Cohen's _d_ measure of effect size, among young-adults completing the Psychology Experiment Building Language (PEBL) neurobehavioral test battery.** Paired _t_-test $^A P < .05$, $^B P < .01$.

Figure 4 depicts the absolute reliability in terms of effect size from the test to the retest for the primary dependent measures with Table S1 also containing secondary indices. Consistent responding (i.e., no significant change) was observed for the number of moves to solve the Tower of London. Slight, but significant ($P \leq .05$) improvements were noted for Digit Span forward and Response Time on the Test of Attentional Vigilance. Significant ($P < .01$) practice effects with a small effect size ($d \geq .2$) were identified for the variability of responding on the Test of Attentional Vigilance, the response pattern on the Iowa Gambling Tasks, the B to A ratio on the Trail Making Test as well as time to complete Part A, and perseverative errors on the Berg Card Sorting task defined according to the Berg criteria. Intermediate ($d \geq .6$) practice effects were identified with increased time on target on the Pursuit Rotor, decreased mean time to solve each Tower of London problem, faster completion of Part B of the Trail Making Test, and heightened accuracy on Time-Wall.

Further analysis on the Iowa Gambling Tasks determined that the amount earned at the end of each session did not appreciably change from the test ($\$1,944.85 \pm 85.04$) to the retest ($2,162.13 \pm 116.03$, $t(67) = 1.59$, $P = .12$; $r(66) = .10$, $P = .40$). However, the number of selections from the disadvantageous decks (1 and 2) decreased 10.3% from the test ($45.6 \pm 1.4$) to the retest ($40.9 \pm 1.8$, $t(67) = 2.66$, $P < .01$, $d = 3.9$; $r(66) = .41$, $P < .0005$).

## DISCUSSION

### Study I: normative behavior & inter-test associations

The principle objective of the first study was to evaluate the utility of a collection of tests from the PEBL battery including convergent and divergent validity. As also noted in the introduction, there are some methodological differences between the PEBL and non-PEBL tasks. The difference between using a stylus versus a computer mouse to track a moving target in the Rotary Pursuit/Pursuit Rotor may not be trivial. The TOVA, but not the TOAV, includes microswitches to record response time which may result in a higher accuracy than may occur without this hardware. Finally, some of these instruments have a prolonged history (*Lezak et al., 2012*) and the dependent measures for some commercial tests (e.g., the WCST and perseverative errors) have evolved over the past six decades (*Berg, 1948*; *Grant & Berg, 1948*; *Heaton et al., 1993*) to be more complex than may be readily apparent based upon reading only the peer-reviewed literature.

The nine measures in this dataset were chosen based on a combination of attributes including assessing distinct neurophysiological substrates (Fig. 1), theoretically meaningful constructs (Table S2), ease and speed of administration, and frequency of use in earlier publications (*Mueller & Piper, 2014*). Admittedly, a potential challenge that even seasoned investigators have encountered with a young-adult "normal" population is that they can quickly and efficiently solve novel problems which may result in ceiling effects (*Yasen et al., 2015*). However, a substantial degree of individual differences were identified on almost all measures (Table 1). The only test where there might be some concern about score distribution would be the points awarded on the Tower of London. A future study (e.g., testing the efficacy of a cognitive enhancing drug) might consider: (1) using alternative measures like completion time; (2) choosing one of the ten other Tower of London already included, e.g., the test contained in *Piper et al. (2012)*, or, as the PEBL code is moderately well documented for those with at least an intermediate level programming ability, to (3) develop their own more challenging test using one of the existing measures as a foundation.

The inter-relationships among tests were characterized to provide additional information regarding validity. For example, indices of attention showed some associations with both motor function and more complex cognitive domains like memory. Overall, the relatively low correlations ($\approx\pm0.3$) between the Trail Making Test, Tower of London, and Berg Card Sorting Test, are congruent with the sub-component specificity of executive function domains (*Miyake et al., 2000*; *Smith et al., 2013*). Similarly, the lack of association of the Iowa Gambling Task with other executive function measures is generally concordant with prior findings (*Buelow & Suhr, 2009*). These findings contribute to a much larger evidence base regarding the validity of measures of the executive functions including volition, planning and decision making, purposive action, and effective performance (*Lezak et al., 2012*; *Smith et al., 2013*).

This dataset also provided an opportunity to examine whether behavior on this battery was sexually dimorphic. Previously, a small ($d = 0.27$) sex difference favoring boys (ages 9–13) was identified on the Pursuit Rotor (*Piper, 2011*). This same pattern was again

observed but was appreciably larger ($d = 0.89$) which raises the possibility that completion of puberty in this young-adult sample may be responsible for augmenting this group difference. On the other hand, in a prior study with 3-dimensional Mental Rotation images and a very similar sample (*Yasen et al., 2015*), sex differences were noted but the effect size was larger ($d = 0.54$) than the present findings ($d = 0.31$). As the PEBL battery currently uses simple 2-dimensional images, image complexity is likely a contributing factor. Sex differences were not obtained on Time-Wall, Berg Card Sorting Tests, Trail-Making, or Tower of London tests which is in-line with earlier findings (*Piper et al., 2012*).

The Berg Card Sorting Test may be the most frequently employed PEBL test in published manuscripts. As both the Berg Card Sorting Test and the Wisconsin Card Sorting Test are based on the same core procedures (*Berg, 1948*; *Grant & Berg, 1948*), these tests appear quite similar from the participant's perspective. However, the sorting rules of *Heaton et al. (1993)* are considerably more complex than those originally developed (*Berg, 1948*; *Grant & Berg, 1948*). The finding that five of the correlations with other tests were significant and of the same magnitude with both and Berg and Heaton rules provides some evidence in support of functional equivalence of these tests.

## Study II: test–retest reliability

The principle objective of the second study was to characterize the test–retest reliability of the PEBL battery with a two-week interval. The correlation between the test and retest is commonly obtained in these types of investigations (*Calamia, Markon & Tranel, 2013*; *Fillmore, 2003*; *Learck, Wallace & Fitzgerald, 2004*; *Lejuez et al., 2005*; *Lezak et al., 2012*; *Woods et al., 2010*). It is also important to be cognizant that the Pearson or the Spearman correlation coefficients may not fully describe the consistency of measurement when the tested participants show an improvement but maintain their relative position in the sample compared to each other. Therefore, a direct comparison between the test and retest scores was also conducted to quantify the extent of any practice effects.

The test–retest correlations were high ($\geq .70$) for the Pursuit Rotor and Test of Attentional Vigilance and moderate ($\geq .30$) for Digit Span and the Berg Card Sorting Test. Some measures on the Iowa Gambling Task and the Tower of London have test–retest reliabilities that were low. It is noteworthy that there is no single value that is uniformly employed as the minimum reliability correlation with some advocates of 0.7 or even 0.8 while others reject the notion of an absolute cut-off (*Calamia, Markon & Tranel, 2013*). In general, an extremely thorough meta-analysis concluded that most tests employed by neuropsychologists have correlations above 0.7 with lower values observed for measures of memory and executive function (*Calamia, Markon & Tranel, 2013*). Many tests that are widely used clinically and for research have test–retest reliabilities that are in the 0.3–0.7 range (*Lowe & Rabbitt, 1998*). Direct comparison with other psychometric reports is difficult because the test–retest intervals and the participant characteristics were dissimilar but they are generally in line with expectations. For example, the present findings ($r = .86$) are slightly higher than what has been reported previously ($r = .75$) for a computerized Pursuit Rotor task (*Fillmore, 2003*). The Test of Variables Attention showed very high

correlations for response time variability ($r = .87$) and response time ($r = .79$) among school-age children with a one-week interval (*Learck, Wallace & Fitzgerald, 2004*) which is identical to the present findings with the TOAV. Healthy (i.e., "normal") adults showed intermediate correlations on both Part A ($r = .46$) and Part B ($r = .44$) of the paper version of the TMT with a 20-week interval (*Matarazzo et al., 1974*) which is lower than the present findings ($r = .61$–$.74$). A mixed (i.e., intact and brain damaged) sample showed moderately high consistency on the Digit Span ($r = .68$) with an eleven month interval (*Dikmen et al., 1999*). Perhaps surprisingly, there is currently very limited reliability data from the non-PEBL computerized versions of the Iowa Gambling Task or the Wisconsin Card Sorting Test for comparative purposes. The pronounced degree of improvement from the test to the retest, whether expressed as the percent change or in terms of Cohen's $d$, are in accord with most earlier findings (*Basso, Bornstein & Lang, 1999*; *Versavel et al., 1997*). Overall, it is important to emphasize that reliability is not an inherent characteristic of a test but instead a value that is influenced by the sample characteristics and the amount of time between the test and retest. The two-week interval would be applicable, for example, to assessing the utility of a cognitive enhancing drug but longer intervals should also be examined in the future.

Some procedural details of many of the PEBL tasks employed in this study are worthy of consideration. The numbers presented in Digit Span and the cards in the Berg Card Sorting Test are selected from a set of stimuli such that the retest will not be identical to the test. The degree of improvement would likely be even larger without this feature. Although not the goal of this report, we suspect that the magnitude of practice effects would be attenuated if alternative versions of tests were employed for the test and the retest. This possibility is already pre-programmed into the Trail Making Test and Tower of London. Similarly, the direction of rotation could be set at clockwise for the test and counterclockwise for the retest if additional study determined equivalent psychometric properties independent of the direction of target rotation. Another strategy that could attenuate practice effects might be to increase the number of trials, particularly on Time-Wall and the Trail Making Test, until asymptotic performance was observed. Further discussion of the varied parameters and the evolution of the Iowa Gambling Task is available elsewhere (*Piper et al., 2015*).

## General discussion

The information obtained regarding the validity and reliability of the majority of PEBL tests is broadly consistent with expectations (*Lezak et al., 2012*; *Lowe & Rabbitt, 1998*) and indicates that these tests warrant further use for basic and clinical research. The overall profile including the distribution of scores, convergent and divergent validity, practice effects being of the anticipated magnitude, and, where applicable, internal consistency, as well as an expanding evidence base (*Mueller & Piper, 2014*), demonstrates that the Pursuit Rotor, Test of Attentional Vigilance, Digit Span, and Trail Making Test are particularly appropriate for inclusion in generalized batteries with participants that are similar to those included in this sample.

One task where the psychometric properties are concerning is the Iowa Gambling Task. An improvement was noted in the response pattern from the test to the retest which is consistent with what would be expected with this executive function test *a priori*. However, the correlation between the test and retest was not even significant when the more conservative statistic (Spearman rho) was examined. Perhaps, in order to attenuate the practice effect, two alternative forms of the Iowa Gambling Task could be developed (e.g., version A where decks 3 and 4 are advantageous and a version B where decks 3 and 4 are disadvantageous). In fact, even more sophisticated alternative forms of the Iowa Gambling Task which vary based on task difficulty are being developed by others (*Xiao et al., 2013*). Another modification which might benefit the test–retest correlation would be to increase the salience of feedback that follows each trial. The feedback was very salient in the original (i.e., non computerized) version of this task in that the experimenter would give or take money after each trial (*Bechara et al., 1994*). Perhaps, the psychometric properties of the PEBL Iowa Gambling Task would be improved if auditory feedback was presented after each trial or there were a fixed interval between trials which would encourage the participant to reflect on their previous selection. These procedural modifications were made for a subsequent study (*Piper et al., 2015*). Overall, additional study is warranted to better appreciate the present findings as there are no long-term test–retest reliability with the non-PEBL computerized Iowa Gambling Task (*Buelow & Suhr, 2009*). However, given the limited evidence for convergent validity or test–retest reliability, prior findings with the PEBL Iowa Gambling Task (*Lipnicki et al., 2009*) may need to be cautiously interpreted.

Three limitations of this report should also be acknowledged. First, the PEBL battery also includes many other indices (e.g., Corsi block tapping test of visuospatial working memory, a Continuous Performance Test of vigilance, a Stroop test of executive functioning). Only a subset of the many PEBL tests were utilized due to time constraints (approximately one-hour of availability for each participant). Future investigations may be designed to focus more narrowly on specific domains (e.g., motor function). Second, a future objective would be to provide further information regarding criterion validity, e.g., by determining the similarities, or differences, between the Test of Variables of Attention and PEBL Test of Attentional Vigilance in neurologically intact and various clinical groups as this information is mostly unavailable for the PEBL tests (although see *Danckert et al., 2012* which utilized the Berg Card Sorting Tests and brain injured patients). Third, different versions of the Tower of London were used in Study I and II which limits inferences across datasets for this measure. More broadly, the psychometric information obtained is most pertinent to the specific tests employed with very specific test parameters. The generalizability of these results to alternative versions of these measures (e.g., a Tower of London with more trials), different sequences of tests, or different (shorter or longer) test batteries will require verification. Fourth, the sample in both studies consisted of young-adult college students, primarily Caucasian and from a middle-class background. There are those that are quite articulate in outlining the limitations of this population (*Henrich, Heine & Norenzayan, 2010*; *Reynolds, 2010*). The data contained in this report

**Table 3 Comparison of computerized neurobehavioral batteries.** Behavioral Assessment and Research System (BARS); Cambridge Neuropsychological Test Automated Battery (CANTAB); Continuous Performance Test (CPT), Maximum (Max); Minimum (Min); Psychology Experiment Building Language (PEBL); Test of Attentional Vigilance (TOAV).

|  | BARS | CANTAB | PEBL |
|---|---|---|---|
| Year developed | 1994 | 1980s | 2003 |
| Origins | Behavior analysis & cognitive psychology | Behavioral neuroscience | Experimental & neuropsychology |
| Philosophy | Working populations with different educations & cultures | Translational, cultural & language independent | Collection of open-source neuropsychological measures |
| Modifiable | No | No | Yes |
| Cost (Min/Max) | $950[a]/$8,450[b] per computer | $1,275[c]/$24,480[d] per computer | Free/free for unlimited computers |
| # Tests | 11 | 25 | >100 |
| Example measures | Finger tapping | Motor screening | Tapping |
|  | Reaction time | Simple reaction time | Rotary pursuit |
|  | CPT | Match to sample | TOAV, CPT |
|  | Digit span | Spatial span | Digit span, Spatial span |
|  | Selective attention | Choice reaction time | Dexterity |
|  | Symbol digit | Stockings of Cambridge | Tower of London |

**Notes.**

[a] One-year preliminary data/student package with 9Button hardware ($450).

[b] Three-year license with hardware.

[c] One-test with one-year license.

[d] All tests for 10 year license.

should just be viewed as an important first step as further investigations with different ages, socioeconomic, and ethnic groups is needed.

There have been several pioneers in the development of new measures which have greatly facilitated our understanding of individual differences in neurobehavioral function (*Lezak et al., 2012*). We feel that the transparency of the PEBL battery extends upon this earlier work and provides an important alternative to commercial tests. In addition, the ability of anyone with a functional computer, independent of their academic degrees, to use PEBL contributes to the democratization of science.

On the other hand, two considerations with PEBL and other similar open-source applications should be acknowledged. First, the flexibility of PEBL also has clear drawbacks in that each investigator can, in theory, modify a test's parameters to meet their own experimental needs. If an investigator reports that they employed a particular test from a specific commercial distributor, there is wide-spread agreement about what this means as many these tests often have only limited modifiability. However, if an investigator changes a PEBL test but fails to make the programming code available, then it is more difficult to critically evaluate research findings. The second potential drawback with PEBL may be ethical. The prohibition against clinical psychologists (*American Psychological Association, 2002*), but not others, making neurobehavioral tests readily available is discussed elsewhere (*Mueller & Piper, 2014*). The accessibility of PEBL to anyone, including psychiatrists, neurologists, or cognitive neuroscientists for research or teaching purposes is consistent with the ethos of science (*Merton, 1979*).

These findings also begin to aid comparisons with other older neurobehavioral test batteries. Table 3 contrasts PEBL with the Behavioral Assessment and Research System (BARS) and, perhaps the current "gold standard" of batteries, the Cambridge Neuropsychological Test Automated Battery (CANTAB) in terms of intellectual origins, the not insignificant differences in price and transparency, and sample tests. The BARS system is based on the behavioral analysis principles of BF Skinner and is designed for testing diverse populations including those with limited education and prior computer experience (*Rohlman et al., 2003*). The CANTAB battery was designed with an emphasis on translating preclinical findings to humans (*Robbins et al., 1994*). Each of these platforms have their own advantages and disadvantages with the strength of PEBL being the number of tests, limited cost, and modifiability.

## CONCLUSION

In closing, this report provides key information regarding the test–retest reliability, convergent and discriminant validity of many commonly employed PEBL tests. Our hope is that thorough, but critical, investigations of the psychometric properties of this novel methodology in normal (present study) and atypical populations will insure that PEBL will continue to be widely used by investigators in basic and applied areas. This will foster further integration between these fields and further advance our understanding of the genetic, biochemical, and neuroanotomical substrates of individual differences in neurocognition.

## ACKNOWLEDGEMENT

The technical assistance of Christopher J. Fox, Vera E. Warren, Hannah Gandsey, Sari N. Matisoff, and Donna M. Nolan is gratefully recognized.

### Funding

This research was supported by the NIDA (L30 DA027582), the Husson University School of Pharmacy, the NL Tarter Trust Award, and NIEHS (T32 ES007060-31A1). The funders had no role in study design, data collection and analysis, decision to publish, or preparation of the manuscript.

### Grant Disclosures

The following grant information was disclosed by the authors:
NIDA: L30 DA027582.
Husson University School of Pharmacy.
NL Tarter Trust Award.
NIEHS: T32 ES007060-31A1.

### Competing Interests

Shane T. Mueller is an Academic Editor for PeerJ. The other authors declare there are no competing interests.

## Author Contributions

- Brian J. Piper conceived and designed the experiments, analyzed the data, wrote the paper, prepared figures and/or tables, reviewed drafts of the paper.
- Shane T. Mueller analyzed the data, contributed reagents/materials/analysis tools, wrote the paper, reviewed drafts of the paper.
- Alexander R. Geerken and Kyle L. Dixon performed the experiments, reviewed drafts of the paper.
- Gregory Kroliczak analyzed the data, contributed reagents/materials/analysis tools, prepared figures and/or tables, reviewed drafts of the paper.
- Reid H.J. Olsen performed the experiments, analyzed the data, contributed reagents/materials/analysis tools, wrote the paper, reviewed drafts of the paper.
- Jeremy K. Miller conceived and designed the experiments, analyzed the data, contributed reagents/materials/analysis tools, wrote the paper, reviewed drafts of the paper.

## Human Ethics

The following information was supplied relating to ethical approvals (i.e., approving body and any reference numbers):

The IRB of WU approved both studies.

## Data Availability

The PEBL scripts for all tasks are available at:

https://github.com/stmueller/pebl-custom/tree/master/Piper-PeerJ-2015b.

## Supplemental Information

Supplemental information for this article can be found online at http://dx.doi.org/10.7717/peerj.1460#supplemental-information.

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
