# Peer review of "Reliability and validity of neurobehavioral function on the Psychology Experimental Building Language test battery in young adults"

_PeerJ, doi:10.7717/peerj.1460_

## Round 0.1 · original submission · Minor Revisions

Thank you for your submission. Both reviewers were generally impressed with the paper and both have made a number of suggestions for minor amendments that will enhance it. Please provide additional detail on the tests reliability and validity in other (non computerised) settings.

·

Basic reporting

Overall, the article is well written. The submission adheres to all PeerJ policies.

The introduction and background sections are exhaustive but not exhausting. The figures are relevant and add to the test.

The authors included the raw data, but I could not find the programming modifications to the PEBL test batteries in the supplemental files, which are mentioned at lines 244-245

Please see my comments in the General Comments for the Author part.

Experimental design

The submission describes an original primary research within the scope of PeerJ.

The design is clearly explicated to the point that the research could be replicated. One minor issue might in the need to replicate the minor tweaks that the authors described at lines 244-245. This only if the tweaks to the source code are not minor and not trivial.

The authors declared that all procedures are consistent with the Declaration of Helinski and were approved by an IRB. The text and the supplied data appear to conform to such declarations.

Please see my comments in the General Comments for the Author part.

Validity of the findings

The data analysis is sound. The authors provided the raw data, so verification is possible.

The discussion and the conclusion of the manuscript are clear and the very few speculations are marked as such.

Please see my comments in the General Comments for the Author part.

Additional comments

The manuscript reports two studies performed to gather first psychometric properties of the Psychology Experimental Building Language (PEBL) test battery. The main objective of the two studies was to report convergent/divergent validity (study I) and test-retest reliability (study II) on the measurements of ten PEBL measures when employed on a group (Study I N=189; Study II N=79) of young-adults aged 18-22. Study II was conducted about two weeks after study I, for assessing test-retest reliability. Table 1 (attached to the review) shows the tests conducted for the two studies in their running order.

Perhaps also due to its open source nature, PEBL is becoming an important tool for studies in the fields related to (neuro)psychology. Furthermore, PEBL has also been used in multidisciplinary research activities, and I have been one of these cases (https://peerj.com/articles/289). A psychometric study of the PEBL batteries was needed.

Overall, I enjoyed reading the manuscript. It is well-written, also for those not completely familiar with the reference discipline. It is short to the point, yet it provides much material for further reading.

The conducted analyses look robust and seem relevant for assessing the validity and the reliability of PEBL, which are the two common directions for measuring psychometric properties. The results, as the authors reported, offer a broadly initial support for the the validity and reliability of PEBL.

This review has a series of comments and questions that I consider minor. Therefore, my suggestions to the editor will be minor revisions.

Although I could download the data related to the two studies, I could not find the programming modifications to the PEBL test batteries in the supplemental files, which are mentioned at lines 244-245. I assume that the modifications are minor, and that the psychometric properties elicited in this manuscript can be transferred to the default PEBL settings.

I kindly ask the authors to state which variation they opted for each included test of the battery. I employed the Tower of London (TOL) test in one of my studies. There were about 10 variations of TOL, two of which were related to Shallice’s test. I assume that a similar situation happens with the other tests of the battery.

The authors state in lines 296-299 that study II employed a more challenging version of TOL to cope with some ceiling effects found in study I. The validity and the reliability measures refer to two different variations of TOL. Does this pose a limitation?

Furthermore, could the authors discuss about to which extent the elicited psychometric properties could be transferred if successive studies employ variations of the tests included in this study’s battery?
The authors state in lines 241-242 that the data from the tapping motor speed test were unavailable. The test is missing from both study I and study II. I wonder if it would be more appropriate to state that the study reports the psychometric properties of 9 PEBL tests.

The study on convergent/divergent validity included ten (nine) PEBL tests. The study on test-retest validity included seven tests. The authors are very clear on this throughout the test. I wonder if the manuscript would improve if it summarized (perhaps in the limitations paragraphs) that the results on validity and reliability are overall not about all ten (nine) test batteries.

At 475, there is a glitch in the text: “task42.”

Finally, I have three comments/questions on the composition of the two batteries. One is a consequence of the other.

The part at 414-417 explains the criteria for inclusion of the chosen tests (Table 1 attached to this review). However, I am missing a discussion regarding the order of the tests and the possible consequences. For example, would changing the order of the tests have an effect to the obtained scores? Consequently, would there be consequences to the reliability and validity scores?

I would like the authors to comment on the consequences of performing 10 tests in about one hour. I recall that for our study, it took the participants about 30mins to listen and read the instructions, open PEBL, and perform all trials of the Shallice’s test ([1, 2, 3] pile heights, 3 disks, and Shallice’s 12 problems). While I do not doubt that it took about 1 hour for performing the 10 tests, I wonder if such a concentration of different tests would have implications on the results and, consequently, on the psychometric properties obtained in the study. The most trivial one that comes to mind would be a fatigue effect.

My third comment is a combination of the previous two. Could the authors comment on how the number of the tests and their ordering may have an implications on the obtained score? For example, the Test of Attentional Vigilance was performed as the tenth test in Study I and as the fourth test in Study II. So we have a difference in terms of time of running, and the number of tests conducted before. What are the consequences for the elicited psychometric properties?

Disclosure: my primary field of study is computer science / software engineering, although my research is multidisciplinary and makes use of psychology theory and measurements. My understanding of some of the constructs measured by PEBL is limited. Therefore, this review may be weakened by such limitations. This concern was disclosed with the academic editor before accepting the peer review request.

Reviewer 2 ·

Basic reporting

On the whole I feel the paper is clearly written and well communicated, though there are a few small sections, such as elements of the results, where I feel some small changes would greatly improve the readability. I also noted that the initialisations of some of the tests do not seem to be clearly stated before use, which may lead to confusion for some readers.

In the introduction I question the inclusion of a detailed description of the key brain areas relating to each study. As, whilst I am aware that these tests are commonly used in neuro-imaging and related research that is by no means their sole function, nor does this paper have any direct relation to that area. As such a feel an in depth description of the relation between these tests and associated brain regions is an unnecessary addition.

There are also a number of minor suggested changes which I have highlighted in the attached PDF comments for the authors to use as they wish.

Experimental design

Again I believe that the design is generally sound, with limitations adequately identified, and addresses an interesting and useful research area. I do however feel that the a few things could be stated a little more clearly. The end of the introduction for example becomes a little busy, and I believe would benefit greatly from a more concise, explicit statement of the research aims and hypotheses. Likewise I feel that the inclusion of a brief statement of the study findings might be a worthwhile addition to the conclusion alongside the hopes for the future.

The selection of the 10 tests is not properly explained until the beginning of the discussion. I would move this into either the end of the introduction or the methods in order to better frame the justification for the inclusions of those particular 10 tests.

A few other points of clarification could also be made in the methods section. For example the reduction in numbers between study 1 and study 2. I would presume that initial consent was collected from all participants in study 1 to attend a two week follow up, and that the subsequent drop in numbers for the test-retest is due to drop outs, but I think it is worth making this clear. Likewise I can make an educated assumption as to why two tests were dropped for study 2, and why only the first two trials were used for the Trial-Making test, but feel the reasons should be stated explicitly.

Validity of the findings

The findings appear to be both robust and interesting, particularly for those who might wish to use these tests in future studies. I also feel the suggestions made by the authors in relation to considerations for specific tests in future research make a useful contribution to the literature.

I would like to see a little more detail in regards to previous findings in regards to the reliability and validity of these tests in other formats, even if, as the authors state, they may not always be directly comparable. As I feel this would frame the paper better within the larger body of literature.

On the whole any speculation is clear, well placed and informed. I do wonder if there is any evidence in the literature for the suggestions made regarding issues with the Iowa Gambling Task that could be added to support it, though I did notice that there appears to already be an investigation of this in review.

Additional comments

As stated previously I have attached a PDF with all of the notes I made during review, some of which may or may not be pertinent to you, but which I hope may be helpful in making some slight changes to improve the clarity and readability of the paper.

Annotated reviews are not available for download in order to protect the identity of reviewers who chose to remain anonymous.

---

## Round 0.2 · accepted · Accept

Thank you for your detailed responses to the concerns of the reviewers.